## Comment

Competence; culture; cultural sensitivity; cross-cultural; developing countries

**Corresponding author:**
Mariana Pinto da Costa;
Email: mariana.pintodacosta@kcl.ac.uk

# Advancing global mental health training: The World Psychiatry Exchange Program 3.0

Mariana Pinto da Costa[1,2,3] , Amine Larnaout[4], Sharad Philip[5], Nicholas Tze Ping Pang[6], Lucija Šenjug Mance[7], Mohammadreza Shalbafan[8,9] , Joan Soler Vidal[10,11] and Tiago Costa[12,13,14]

[1]Institute of Psychiatry Psychology & Neuroscience, King's College London, UK; [2]Institute of Biomedical Sciences Abel Salazar, University of Porto, Portugal; [3]South London and Maudsley NHS Foundation Trust, UK; [4]Razi Hopsital, Faculty of Medicine, University of Tunis El Manar, Tunisia; [5]Department of Psychiatry, Clinical Neurosciencs and Addiction Medicine, All India Institute of Medical Sciences, Guwahati Assam, India; [6]Universiti Malaysia Sabah, Malaysia; [7]University Psychiatric Hospital Sveti Ivan, Croatia; [8]Mental Health Research Center, Psychosocial Health Research Institute (PHRI), Department of Psychiatry, School of Medicine, Iran University of Medical Sciences, Tehran, Iran; [9]Brain and Cognition Clinic, Institute for Cognitive Sciences Studies, Tehran, Iran; [10]Hospital Nostra Senyora de Meritxell, Servei Andorrà Atenció Sanitària, Escaldes, Andorra; [11]FIDMAG Research Foundation, Germanes Hospitalàries, CIBERSAM-ISCIII, Spain; [12]Newcastle University Translational and Clinical Research Institute, UK; [13]Cumbria Northumberland Tyne and Wear NHS Foundation Trust, UK and [14]National Institute for Health and Care Research (NIHR) Newcastle Biomedical Research Centre, UK

## Abstract

Global migration is reshaping mental healthcare, creating challenges and opportunities that demand intercultural dialogue. In 2021 the World Psychiatry Exchange Program was launched under the auspices of the World Psychiatric Association to promote global collaboration and mutual learning. Its third call for applications, opened in October 2023, received 162 applications from 68 individuals, more than double the number from the second edition. Applicants represented a diverse geographical distribution (Asia 58.8%, Africa 22.1%, Europe 10.3%, South America 5.9% and North America 2.9%), with ages ranging from 25 to 52 years (mean 34 years). Just over half (53%) were early career psychiatrists within 7 years of specialising, while 47% were psychiatry trainees. Following a competitive selection process, 15 psychiatrists undertook exchanges in 2024 across Europe, Africa and Asia, with placements in Croatia, India, Iran, Malaysia, Tunisia, Spain and the United Kingdom. Evaluation data showed consistently positive feedback: 82% strongly endorsed the clarity and ease of the application process, and all participants reported feeling well supported by local coordinators. As psychiatry responds to global demographic change, investment in intercultural competencies and flexible training pathways is essential. The psychiatrist of the future is a global psychiatrist, equipped to deliver care, education, and leadership globally.

## Impact statement

Global migration and increasing multicultural societies are transforming the practice of psychiatry. This reality makes it essential for psychiatrists to develop culturally sensitive skills and be able to work across borders. This article highlights how the World Psychiatry Exchange Program, by offering short, supervised exchanges delivered in-person or online, can help early career psychiatrists build these competencies.

The World Psychiatry Exchange Program offers a practical and scalable model. It pairs early career psychiatrists with host teams across settings. This exposes participants to culturally diverse expressions of distress, and helps them develop skills to communicate more clearly with patients and families, adapting evidence-based care to different local contexts. It is also an opportunity to start mentorship relationships which can be sustained remotely, and for participants to build the confidence and leadership skills needed to champion inclusive practice at home.

The broader impact extends beyond individual participants. The programme shows that supporting professional mobility and digital collaboration strengthens the capacity of host services to address culturally diverse mental health needs. It provides a model for integrating global perspectives into clinical practice, education and leadership. By identifying common clinical themes across cultures and exploring how diagnostic frameworks can be applied in culturally sensitive ways, the programme contributes to developing cross-cultural psychiatry as a discipline.

Ultimately, the World Psychiatry Exchange Program demonstrates that international collaboration and mobility are essential for preparing psychiatrists to work in a connected, multicultural world. It highlights the need for institutions and professional organisations to invest in flexible training pathways, digital infrastructure and intercultural competence. This article underscores that the psychiatrist of the future is not only clinically skilled but also globally aware, adaptable and capable of leading and delivering mental healthcare in diverse cultural contexts.

## Comment

The world is becoming increasingly pluralistic, multilingual and multicultural, driven largely part by rising in global migration. These demographic changes present both challenges and opportunities for mental healthcare. As people move, they carry distinct cultural frameworks; for example, differing beliefs about the causes of illness, preferred coping strategies or attitudes towards professional help. These differences can lead to a disconnect between patient's lived experiences and a clinician's culturally bound diagnostic tools and treatment approaches. Mental health conditions do not arise in isolation but are shaped by culture, language and societal norms. In this context, understanding this variability is not just helpful, it is essential.

Mobility of mental health professionals, both physically and digitally, has therefore become crucial. International exchange programmes, digital tools and remote collaborations are reshaping how mental health professionals learn, treat and lead across borders (Pinto da Costa & Sartorius, 2022). Intercultural dialogue – the reciprocal exchanges of perspectives between clinicians and patients, or among professionals from different cultural traditions – strengthens clinical judgement. Early career psychiatrists exposed to different systems report a greater ability to recognise and treat mental illnesses across cultural contexts (Ben Said et al., 2023; Daoud et al., 2024). With rising global migration and diasporas, such knowledge is increasingly invaluable (Pinto da Costa et al., 2017). Psychiatry, therefore, cannot remain domestically anchored; it must be globally aware, responsive and mobile. Such global awareness is essential for effective care and for guiding leadership in psychiatric practice.

Psychiatrists around the world commonly rely on standardised diagnostic frameworks such as the International Statistical Classification of Diseases and Related Health Problems (ICD) or the Diagnostic and Statistical Manual of Mental Disorders (DSM) (Pinto da Costa et al., 2021). They are validated and useful frameworks but cultural variation exposes their limitations and highlights the need for careful interpretation and clinical acumen (Lewis-Fernandez & Aggarwal, 2013). For instance, for the diagnosis of mood disorders, social withdrawal, impaired functioning and persistent distress are recognised across contexts, while somatic presentations and the content of negative cognitions may have significant cultural variation (Simon et al., 1999). The shared features can serve as anchors for common ground, while culturally specific expressions should be considered to ensure accuracy and sensitivity. In the World Psychiatry Exchange Program, initiatives such as collaborative case discussions help highlight commonalities and the value of culturally specific symptoms, helping participants to refine their use of DSM and ICD criteria in culturally responsive ways.

The World Psychiatry Exchange Program exemplifies how international engagement is reshaping psychiatric education and collaboration (Pinto da Costa et al., 2025; Pinto da Costa et al., 2023). Launched in 2021, the programme has supported early career psychiatrists with placements across all continents: Asia, Europe, Africa, Oceania and the Americas. These exchanges (both in-person and remote) offer unique opportunities to learn from different clinical systems, research models and therapeutic approaches. Faculty members from the host departments report very positive experiences of this exchange programme, considering it invaluable for the development of the next generation of psychiatrists and psychiatric trainees (Kamalzadeh et al., 2023).

The third call for applications for the World Psychiatry Exchange Program opened in October 2023 with placements available in various parts of the world, including Brazil, Croatia, France, India, Iran, Malaysia, Nigeria, Spain, Tunisia and the United Kingdom (UK). In this third edition, the total number of individual applications more than doubled to 162, compared to the 66 from the second edition (Pinto da Costa et al., 2025). In total, 68 people applied, generating 162 applications, as some applied to multiple placements. Of the applicants, 61% were female, with ages ranging from 25 to 52 years (mean 34 years), from Asia ($N = 40$, 58.8%), Africa ($N = 15$, 22.1%), Europe ($N = 7$, 10.3%), South America ($N = 4$, 5.9%) and North America ($N = 2$, 2.9%). The candidates were either early career psychiatrists within 7 years of specialising in psychiatry ($N = 36$, 53%) or psychiatry trainees ($N = 32$, 47%). Following a competitive selection process, 15 early career psychiatrists completed exchanges in 2024, in Europe ($N = 7$), Africa ($N = 3$) and Asia ($N = 5$). Most participants (57%) were placed in their first-choice location and 21% in their second choice. Placement sites included Croatia, India, Iran, Malaysia, Tunisia, Spain and the UK.

Feedback was overwhelmingly positive. Nine participants (82%) "completely agreed" that the application process was easy, and 10 participants (91%) that outcomes were communicated promptly. All participants "completely agreed" that communication with the local coordinator was easy, and that they felt supported throughout the exchange programme. The only item with an average score under 4 (out of 5) was the "ease of arranging a Visa to travel," highlighting the importance of offering remote options for inclusivity.

Remote participation has proven to be a powerful opportunity (Naskar et al., 2022) to address barriers such as visa restrictions and travel costs. Participants highlighted the value of remote learning environments that enable clinical discussions and academic exchange, reducing geographic limitations while preserving the richness of global exposure. One participant noted "*I truly appreciated our discussions on diagnostics and psychopathology, and it was fascinating to learn more about English psychiatry.*" Technology allows psychiatrists to provide consultations, monitor treatments and train peers without being physically present. Another participant reflected that the exchange allowed them the "*honing my clinical skills through collaborative learning with colleagues… fostering cultural competence and communication skills.*"

Global mobility also shapes careers. Early career psychiatrists involved in the World Psychiatry Exchange Program frequently publish together about their cross-cultural insights during the exchanges, build networks, conduct research together and launch new initiatives in collaboration with peers from other countries (Belinati Loureiro et al., 2023; Ben Said et al., 2023; Daoud et al., 2024; Naskar et al., 2022). One participant said: "*the clinical experience I gained … has made profound impact on my future career development, as I plan to implement acquired knowledge in my institution and try to organize such services in [my country"].*" Another wrote: "*The most rewarding aspect of my experience at [the] Hospital was forming meaningful friendships with the people there. These connections have enriched my personal and professional journey, making the program truly memorable and valuable,*" showing the personal and professional benefits of international engagement. Other participant shared: "*I am incredibly grateful to the association for giving me this opportunity.*"

Several participants have gone on to host early career psychiatrists themselves as part of this programme, creating a flywheel effect of international collaborations and global mentorship. This cross-border dynamic reinforces psychiatry as a globally connected discipline. Supporting mobility, through exchanges and digital collaborations, ensures that professional development is not confined to

local contexts. Psychiatrists can train, practice and grow professionally across borders. To fully realise this vision, institutions should support flexible training pathways, provide financial or in-kind support for travel and accommodation, invest in digital infrastructure and promote multilingual, multicultural competencies in psychiatric education.

The momentum behind global mental health is clear. As the World Psychiatric Association prepares for further programme expansions, the message is unmistakable: the future psychiatrist is a global psychiatrist.

**Open peer review.** To view the open peer review materials for this article, please visit http://doi.org/10.1017/gmh.2025.10072.

**Acknowledgements.** Thank you to all the other colleagues who offered to host early career psychiatrists in their institutions in the first four editions of the World Psychiatry Exchange Program: Paul Fung (New Zealand), Camille Noël (Belgium), Gary Cheung (New Zealand), Irena Rojnić Palavra (Croatia), Jasmine Ma (Nepal), Karla Lašakrin (Croatia), Kopal Rohatgi (India), Margaret Isioma Ojeahere (Nigeria), Mariana Paim Santos (Brazil), Natko Geres (Croatia), Paula Lopez Garcia (UK), Ramdas Ransing (India), Rick Wolthusen (USA), Rodrigo Ramalho (Australia), Anne Rozencwajg (France) and Vinicius Belinati Loureiro (Brazil). Thank you to all the World Psychiatry Exchange Program participants for their valuable feedback and many contributions to host organisations over the years.

**Author contribution.** Mariana Pinto da Costa wrote the first draft of the manuscript; all authors reviewed and approved the final version. Tiago Costa designed the graphical abstract.

**Competing interests.** Mariana Pinto da Costa has been the founder of the World Psychiatry Exchange Program. Mariana Pinto da Costa, Amine Larnaout, Sharad Philip, Nicholas Pang Tze Ping, Lucija Šenjug Mance, Mohammadreza Shalbafan and Joan Soler Vidal have hosted early career psychiatrists in their institutions for the World Psychiatry Exchange Program in 2024. Tiago Costa has designed and developed the website and online workflows for the World Psychiatry Exchange Program.

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
