## [Reviewer Report]

This is well written however it can be strengthened by unpacking some of the terms used. For instance the sentence starting (line 9) “clinicians' culturally informed diagnostic tools and therapeutic approaches..” could go further and give an example if not in the abstract maybe in the main body of the comment. Similarly line 13 “ intercultural dialogue”.....unpacking this would help the reader to appreciate the article more. Should line 23 starting “ from 68 individuals...” be 68 countries?“ The paragraph from line 36 through to 44 is very specific about challenges ( visa acquisition) but no specific examples of the positives are highlighted, maybe one or two very specific examples to support the line starting ” all participants reported feeling well supported.....“ Line 47 ”participants frequently publish together...." an example of some of the publications and topics would be helpful. Again this could be in the main body of the text and not in the abstract.

Comment section

As indicated above, there is need to unpack some of the terms such as “cultural framework” and “intercultural dialogue”

Furthermore, with psychiatrists globally relying on the DSM 5 or ICD diagnostic framework, it would be helpful to see how through a cultural lens one navigates through these frameworks. For instance, what are some of the common themes perceptions that seem to be cross-cutting and can be leveraged to strengthen cross-cultural psychiatry? Are some of these themes reflected in joint initiatives between the participants? If yes maybe sharing some of these examples and practical steps would further strengthen the article.

---

## [Editor Report]

Thank you for addressing the issues raised by the reviewers. The manuscript is suitable for publication.